# Linearly-evolved Transformer for Pan-sharpening

Junming Hou*
Southeast University
Nanjing, China
junming_hou@seu.edu.cn

Zihan Cao*
University of Electronic Science and
Technology of China
Chengdu, China
iamzihan666@gmail.com

Naishan Zheng
University of Science and Technology
of China
Hefei, China
nszheng@mail.ustc.edu.cn

Xuan Li
Southeast University
Nanjing, China
xuanli2003@seu.edu.cn

Xiaoyu Chen
Southeast University
Nanjing, China
213214058@seu.edu.cn

Xinyang Liu
Hong Kong Polytechnic University
Hong Kong, China
codex.lxy@gmail.com

Xiaofeng Cong
Southeast University
Nanjing, China
cxf_svip@163.com

Danfeng Hong†
Aerospace Information Research
Institute, Chinese Academy of
Sciences
Beijing, China
hongdf@aircas.ac.cn

Man Zhou†
University of Science and Technology
of China
Hefei, China
manman@mail.ustc.edu.cn

## Abstract

Vision transformer family has dominated the satellite pan-sharpening field driven by the global-wise spatial information modeling mechanism from the core self-attention ingredient. The standard modeling rules within these promising pan-sharpening methods are to roughly stack the transformer variants in a cascaded manner. Despite the remarkable advancement, their success may be at the huge cost of model parameters and FLOPs, thus preventing its application over low-resource satellites. To address this challenge between favorable performance and expensive computation, we tailor an efficient linearly-evolved transformer variant and employ it to construct a lightweight pan-sharpening framework. In detail, we deepen into the popular cascaded transformer modeling with cutting-edge methods and develop the alternative 1-order linearly-evolved transformer variant with the 1-dimensional linear convolution chain to achieve the same function. In this way, our proposed method is capable of benefiting the cascaded modeling rule while achieving favorable performance in the efficient manner. Extensive experiments over multiple satellite datasets suggest that our proposed method achieves competitive performance against other state-of-the-art with fewer computational resources. Further, the consistently favorable performance has been verified over the hyper-spectral image fusion task. Our main focus is to provide

an alternative global modeling framework with an efficient structure. The code is publicly available at https://github.com/coder-JMHou/LFormer.

## CCS Concepts

• **Computing methodologies** → **Computer vision problems**.

## Keywords

Pan-sharpening, Transformer, Image fusion

**ACM Reference Format:**
Junming Hou, Zihan Cao, Naishan Zheng, Xuan Li, Xiaoyu Chen, Xinyang Liu, Xiaofeng Cong, Danfeng Hong, and Man Zhou. 2024. Linearly-evolved Transformer for Pan-sharpening. In *Proceedings of the 32nd ACM International Conference on Multimedia (MM '24), October 28-November 1, 2024, Melbourne, VIC, Australia.* ACM, New York, NY, USA, 9 pages. https://doi.org/10.1145/3664647.3680979

*Equal Contribution.
†Corresponding authors.

## 1 Introduction

The proliferation of remote sensors has made explosive satellite imagery accessible across diverse domains such as military systems, environmental monitoring, and mapping services [7, 22]. Given the inherent physical constraints, satellites typically employ multi-spectral (MS) and panchromatic (PAN) sensors to capture the complementary information concurrently. Specifically, MS images exhibit superior spectral resolution but limited spatial resolution, whereas PAN images provide abundant spatial details but lack spectral resolution. Consequently, the fusion of MS and PAN images through Pan-sharpening techniques has garnered escalating interest from the image processing and remote sensing communities, enabling the generation of images with enhanced spectral and spatial resolutions.

In recent years, convolutional neural networks (CNN) have made significant strides in the satellite pan-sharpening field, surpassing traditional optimization pan-sharpening methods by a substantial margin, thanks to their powerful learning capability. However,

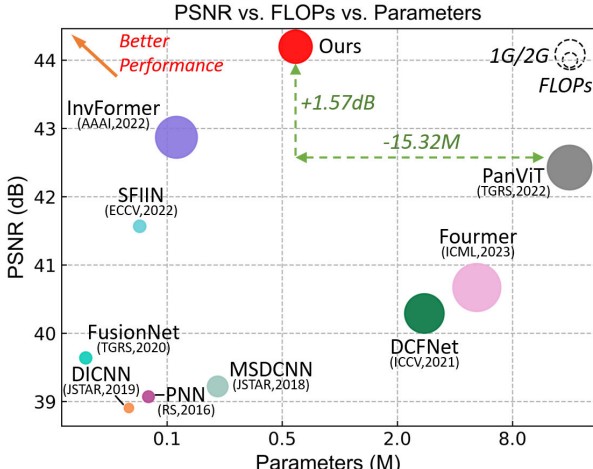

**Figure 1: The comparison of PSNR and computational overhead between our model and other cutting-edge techniques. Notably, the Parameters axis is depicted using a logarithmic scale with a base of 2 for clear illustration. It is evident that our method showcases the promising performance-efficiency balance compared to other approaches.**

the landscape has recently been disrupted by the emergence of the vision transformer family, which challenges the dominance of CNNs by leveraging global-wise spatial modeling based on dot-product self-attention. Among the transformer-based methods, INNformer [40] stands out as a representative approach, employing a multi-modal transformer to capture long-range cross-modality relationships and outperforming previous CNN-based methods. Since then, a multitude of explosive complex transformer variants-equipped pan-sharpening architectures have emerged and solidified their position at the forefront [2, 10, 13]. Notably, these promising transformer-based pan-sharpening architectures generally adhere to a cascaded stacking of transformer variants as a common modeling rule. As displayed in Figure 1, despite their remarkable progress, the success of these approaches often comes at the expense of increased model parameters and floating-point operations (FLOPs), limiting their applicability in low-resource satellite scenarios. To tackle the aforementioned challenge of balancing high performance with substantial computation costs, we delve into the origins of the computation cost, identifying the dot-product self-attention mechanism as a major contributor. In our investigation, we delve into the underpinnings of self-attention and inquire whether an alternative 1-order modeling mechanism could replace the current transformer chain in a more computationally efficient manner. By exploring this avenue, we aim to find a solution that maintains performance while mitigating the resource-intensive nature of the transformer architecture.

Building upon the aforementioned principle, we delve further into the widely adopted cascaded transformer modeling approach used in state-of-the-art methods and design a linearly-evolved transformer as illustrated in Figure 2. This revelation leads us to develop an alternative 1-order linearly-evolved transformer variant

using a chain of 1-dimensional linear convolutions. By leveraging this design, we construct a lightweight pan-sharpening framework that relies on a well-designed, simple yet efficient linearly-evolved transformer. This framework aims to strike a balance between computational efficiency and performance in pan-sharpening tasks. By adopting this design, our proposed method harnesses the advantages of the cascaded modeling rule while achieving impressive performance in a computationally efficient manner. Through extensive experimentation on various satellite datasets, we have demonstrated that our method delivers competitive performance compared to other state-of-the-art approaches while utilizing fewer computational resources. Our primary objective is to provide an alternative global modeling framework with an efficient structure, prioritizing both performance and resource efficiency. The work's contributions are summarized as follows:

- We introduce a novel, lightweight, and efficient pan-sharpening framework that achieves competitive performance while reducing computation costs compared to state-of-the-art pan-sharpening methods.

- We uncover the 1-order principle of self-attention and propose a linearly-evolved transformer chain that replaces the common modeling rule of N-cascaded transformer chains with a feasible approach utilizing 1-transformer and N-1 1-dimensional convolutions to achieve the same function.

- The proposed linearly-evolved transformer provides an effective alternative for global modeling, offering significant potential for designing efficient models.

## 2 Related Works

### 2.1 Pan-sharpening

Existing pan-sharpening methods can be roughly divided into four types: component-substitution (CS)-, multi-resolution analysis (MRA)-, variational optimization (VO), and deep learning-based methods [22]. Among them, the first three categories are also classified as traditional methods. The fused results of CS-based approaches often exhibit significant spectral distortion [3, 14], while the products from MRA-based methods suffer from spatial distortion despite they show superior spectral quality in comparison to CS-based approaches [1, 15, 24]. VO-based methods generate the image with desirable spatial-spectral preservation conditioned on the heavy computational burden [6, 32, 33]. Recently, deep learning-based methods have dominated the pan-sharpening field. The pioneering work only consists of three convolution layers, while achieving a competitive result compared with traditional methods [21]. Subsequently, Yang *et al.* propose the first deeper CNN for pan-sharpening [35]. Since then, more complicated network architectures have been designed for pan-sharpening, showing significant performance gains while leading to high computational and memory footprint [17, 26, 43].

### 2.2 Transformer Based Deep Learning Methods

Very recently, the vision transformer family has dominated the satellite pan-sharpening field. In pioneering works, however, researchers roughly employ the cascaded vision transformer designs

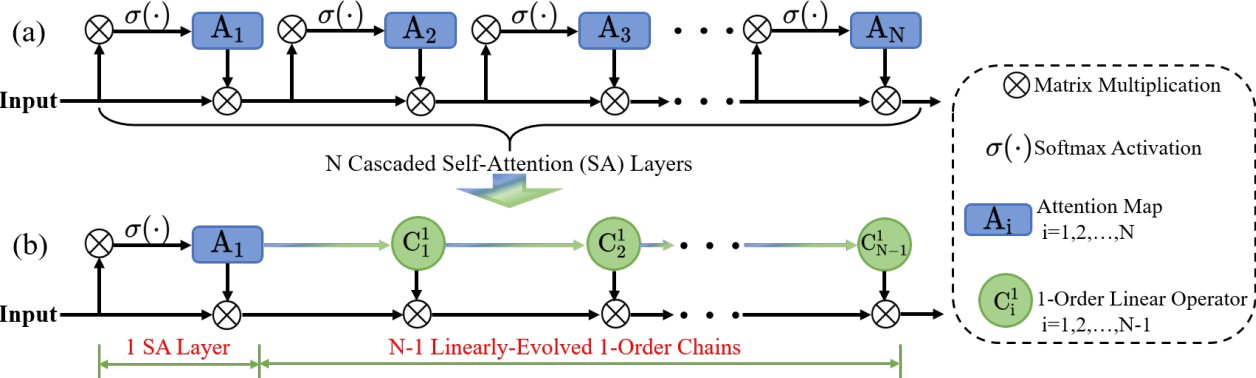

**Figure 2: The comparison between the prior cascaded self-attention designs within transformer and our proposed linearly-evolved mechanism. In this way, our linearly-evolved design is capable of inheriting the merits of a cascaded manner with the huge computation cost reduction.**

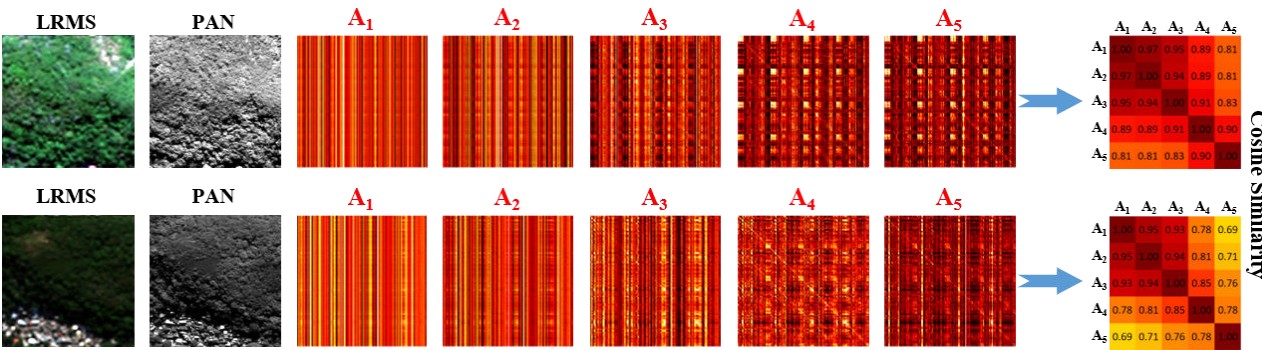

**Figure 3: Attention similarity. Illustration of attention maps across different layers from a cascaded vision transformer (ViT) architecture [23] on the WorldView-3 testing dataset.** $A_i(i = 1, \cdots, 5)$ **denotes the attention map from the** *i*-**th ViT block. The cosine similarity analysis reveals the high similarity among attention maps from various ViT blocks, resulting in feature representation redundancy and unnecessary computations. This motivates us to explore a more efficient alternative solution for effectively modeling feature dependencies, improving pan-sharpening performance, yet reducing the computational overhead.**

to the pan-sharpening problem [10, 23], which ignore some task-related characteristics, *e.g.,* the difference between input source images. Afterward, more task-specific transformers are designed for pan-sharpening. For example, Bandara *et al.* [2] propose a textural and spectral feature fusion transformer for pan-sharpening, dubbed HyperTransformer, whose queries and keys are provided by the features of LR-HSI and PAN, respectively; Zhou *et al.* [40] first introduce transformer and invertible neural network into the pan-sharpening field, in which the PAN and MS features are formulated as queries and keys to encourage joint feature learning across two modalities. Despite the remarkable advancement, existing transformer-based pan-sharpening models suffer from huge network parameters and FLOPs owing to the repetitive self-attention calculation, which heavily hinders their application over low-resource satellites. Moreover, such dense self-attention computing within existing paradigms often leads to high representation redundancy as revealed by the highly similar attention maps across different layers shown in Figure 3.

Lately, much work has endeavored to develop efficient attention. Lu *et al.* propose a Softmax-free transformer, in which the Gaussian kernel function is used to replace the dot-product similarity [20]. Zhai *et al.* explore an attention-free transformer, which eliminates the need for dot product self-attention [38]. Liu *et al.* propose Eco-Former, a new binarization paradigm, which maps the original queries and keys into low-dimensional binary codes in Hamming space [16]. Guo *et al.* develop a novel external attention with linear complexity, which is implemented through two cascaded linear layers and two normalization layers [8]. Venkataramanan *et al.* propose reusing the output of self-attention blocks to reduce unnecessary computations [27]. Yang *et al.* replace self-attention with a novel focal modulation module for modeling token interactions in vision [36]. In general, these improvements mainly focus on eliminating or replacing the dot product in the self-attention module. More importantly, most of them are tailored toward high-level vision tasks, such as image classification and image segmentation, with limited exploration in pixel-level tasks.

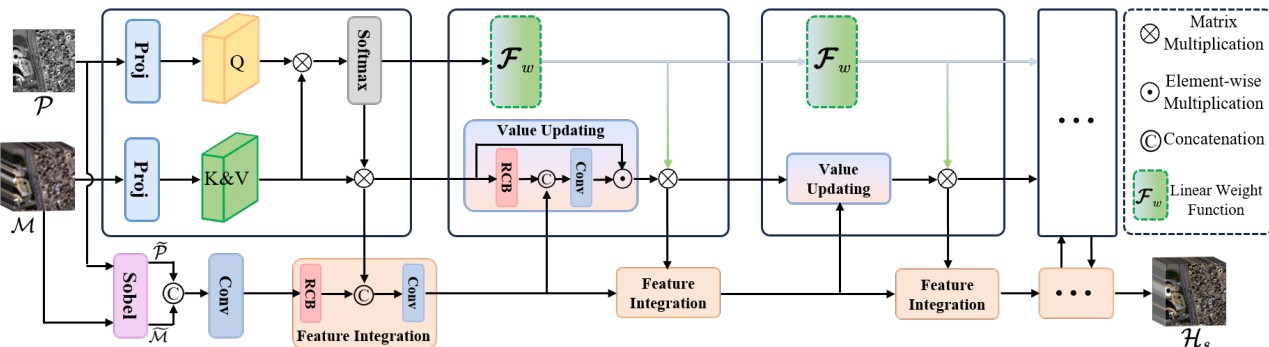

**Figure 4: Overall architecture of the proposed lightweight pan-sharpening framework. LFormer is the core design of our model, where self-attention is replaced by a novel linearly-evolved attention. Sobel and RCB denote Sobel operator and residual convolution block. $\mathcal{F}_w$ represents the linear weight function used for evolving the attention weights. For simplicity, herein, we opt for a straightforward 1-D convolution operator followed by the Softmax function to accomplish this fundamental design.**

## 3 Proposed Method

We first summarize the overall framework, and then revisit the modeling principle of self-attention within the traditional transformer and provide the details of the alternative linearly-evolved transformer chain, which is the core design of our work.

### 3.1 Overall Framework

Figure 4 outlines the overall architecture of the proposed method, which consists of two branches. Given an up-sampled MS image $\mathcal{M} \in \mathbb{R}^{H \times W \times c}$ and PAN image $\mathcal{P} \in \mathbb{R}^{H \times W \times 1}$, the upper branch applies an input projection block to extract their shallow features. While the below one initially employs a high-pass filter to obtain their high-frequency details, denoted as $\widetilde{\mathcal{M}} \in \mathbb{R}^{H \times W \times c}$ and $\widetilde{\mathcal{P}} \in \mathbb{R}^{H \times W \times 1}$, which are further projected into the feature space. Then, we conduct the cross-attention computation between MS and PAN features, yielding the long-range dependency feature representation, which is further interacted with the extracted high-frequency features. Next, we conduct several core building modules **LFormer** coupled with feature integration blocks to obtain the informative features, and then combine with the input $\mathcal{M} \in \mathbb{R}^{H \times W \times c}$ to reconstruct a high-resolution MS image $\mathcal{H}_s \in \mathbb{R}^{H \times W \times c}$. Briefly, our method can be expressed as follows:

$$\mathcal{H}_s = \text{LFormer}\left\{\Phi(\mathcal{M}, \mathcal{P}), \Psi(\widetilde{\mathcal{M}}, \widetilde{\mathcal{P}})\right\} + \mathcal{M}, \quad (1)$$

where $\Phi(\cdot)$ and $\Psi(\cdot)$ involve extracting the initial long-range features and the high-frequency detail information of the source images, respectively. **LFormer** $\{\cdot\}$ denotes the mapping function of the proposed linearly-evolved transformer chain.

### 3.2 The Underlying Principle of Linearly-evolved Transformer

**Revisiting the Traditional Multi-head Self-attention.** Given an input feature $\mathcal{X} \in \mathbb{R}^{H \times W \times C}$ flatten to $HW \times C$, where H and W represent the height and width, respectively, while C is the channel number. Vision Transformer often applies the self-attention module to deal with the input feature, which can be mathematically

formulated as follows:

$$\mathcal{Y} = \text{Softmax}(\frac{QK^T}{\sqrt{d}})V = A \otimes V, \quad (2)$$

where $Q, K$ and $V \in \mathbb{R}^{HW \times C}$ are embedded from $\mathcal{X}$, $\mathbf{A} \in \mathbb{R}^{HW \times HW}$ denotes attention map calculated by $Q, K$. $\otimes$ means the matrix multiplication. $\mathcal{Y}$ is the output of the self-attention module. Furthermore, we can obtain the following expression:

$$\mathbf{a}_{i,j} = \frac{\exp(\mathbf{q}_i \mathbf{k}_j)}{\sum_{m=0}^{HW-1} \exp(\mathbf{q}_i \mathbf{k}_m)}, \quad (3)$$

where $\mathbf{q}_i \in Q$, $\mathbf{k}_j \in K$, $\mathbf{a}_{i,j} \in A$. In terms of $\mathcal{Y} \in \mathbb{R}^{HW \times C}$, we can obtain the following formula:

$$\mathbf{y}_i = \sum_{m=0}^{HW-1} \mathbf{a}_{i,m} \mathbf{v}_m,$$
$$\mathcal{Y} = \mathcal{O}_1(V), \quad (4)$$

where $\mathcal{O}_1(\cdot)$ indicates the 1-order weights.

**Delving into the Modeling Rule of Cascaded Transformer Chain.** Very recently, the vision transformer family has dominated the satellite pan-sharpening field driven by the global-wise spatial information modeling mechanism from the core self-attention ingredient. The common modeling rules within these promising pan-sharpening methods are to roughly stack the transformer variants in a cascaded manner, which can be formulated as follows:

$$\mathcal{X} \to (Q_1, K_1, V_1) \to (Q_2, K_2, V_2) \cdots \to (Q_r, K_r, V_r)$$
$$\cdots \to (Q_N, K_N, V_N) \quad (5)$$
$$\mathcal{X} \to \mathcal{O}_1(V_1) \to \mathcal{Y}_1 \to \mathcal{O}_1(V_2) \to \mathcal{Y}_2 \cdots \to \mathcal{O}_1(V_r)$$
$$\to \mathcal{Y}_r \cdots \to \mathcal{O}_1(V_N) \to \mathcal{Y}_N$$

Taking the adjacent two steps of the above Markov's chain, for example, we can summarize it as

$$\mathcal{Y}_r = \mathcal{O}_1(V_r) \propto \mathcal{O}_1(\mathcal{O}_1(V_{r-1})). \quad (6)$$

We give out the proof next. Similarly, standing on the output $\mathcal{Y}_r$, the r + 1 step performs the dot-product self-attention as

$$(Q_{r+1}, K_{r+1}, V_{r+1}) \leftarrow (W_q^{r+1} \mathcal{Y}_r, W_k^{r+1} \mathcal{Y}_r, W_v^{r+1} \mathcal{Y}_r),$$

$$\mathcal{Y}_{r+1} = \textbf{Softmax}(\frac{Q_{r+1} K_{r+1}^T}{\sqrt{d}}) V_{r+1} \tag{7}$$
$$= A_{r+1} \otimes V_{r+1} \propto \mathcal{O}_1(V_{r+1}) \propto \mathcal{O}_1(\mathcal{Y}_r),$$

where A is the self-attention map, To emphasize, the A is independent as V. Based on the above principle, we can deduce as

$$\mathcal{Y}_{r+1} \propto \mathcal{O}_1(V_{r+1}) \propto \mathcal{O}_1(\mathcal{Y}_r) \propto \mathcal{O}_1(V_{r-1}) \propto$$

$$\mathcal{O}_1(\mathcal{Y}_{r-2}) \dots \mathcal{O}_1(V_1) \propto \textbf{Softmax}(\frac{Q_1 K_1^T}{\sqrt{d}}) V_1 \tag{8}$$

**Linearly-evolved Transformer.** Therefore, the above calculation can be simplified with any form of the 1-order weight function. Referring to the above calculating,

$$\mathcal{X} \rightarrow (Q_1, K_1, V_1) \rightarrow (Q_2, K_2, V_2) \cdots \rightarrow (Q_r, K_r, V_r)$$
$$\dots \rightarrow (Q_N, K_N, V_N), \tag{9}$$

and it can be modeled as follows:

$$\mathcal{X} \rightarrow (Q_1, K_1, V_1) \rightarrow (A_1, V_1) \rightarrow (Q_2, K_2, V_2)$$
$$\rightarrow (A_2, V_2) \cdots \rightarrow (Q_N, K_N, V_N)$$
$$\mathcal{X} \rightarrow (Q_1, K_1, V_1) \rightarrow (A_1, V_1) \rightarrow (C_1^1 * A_1, V_2) \tag{10}$$
$$\dots \rightarrow (C_{N-1}^1 * A_{N-1}, V_N)$$

In our study, we utilize the basic 1-dimensional convolution $C_i^1$ ($i = 1, \cdots, N-1$) with the kernel size of $1 \times k$ to address 1-order functions, where $*$ denotes the convolution operation. To emphasize, the complexity of the previous self-attention mechanism A is quadratic. In contrast, our 1-dimensional convolution design $C_i^1$ exhibits linear complexity. It can be seen that our design reduces the complexity by 1 order of magnitude.

## 3.3 Architecture Details

As described in Equation 1, our proposed framework includes two fundamental components: the **LFormer**$(\cdot)$ branch and the feature integration branch.
**Flow of the LFormer Branch.** In our proposed LFormer branch, we first project the PAN image $\mathcal{P}$ and the up-sampled MS image $\mathcal{M}$ into the feature space, denoted as $\mathcal{F}_\mathcal{P}$ and $\mathcal{F}_\mathcal{M}$, through convolution layers with non-linear activation, formulated as follows:

$$\mathcal{F}_\mathcal{P} = \text{Proj}(\mathcal{P}), \quad \mathcal{F}_\mathcal{M} = \text{Proj}(\mathcal{M}), \tag{11}$$

where Proj$(\cdot)$ denotes the convolution layers with reshape operation. Next, we conduct the cross-attention computation to capture the long-range dependency representation between PAN and MS modalities. Targeting at pan-sharpening, we take $\mathcal{F}_\mathcal{P}$ as query, while $\mathcal{F}_\mathcal{M}$ is used as key and value, written as following form:

$$A_1 = \text{Softmax}(\frac{\mathcal{F}_\mathcal{P} \mathcal{F}_\mathcal{M}^T}{\sqrt{d}}), \quad \mathcal{F}_1^g = A_1 \otimes \mathcal{F}_\mathcal{M}, \tag{12}$$

where $A_1$ denotes the calculated attention map, $\mathcal{F}_1^g$ is the output of the cross-attention module. After that, we employ $N$ **LFormer**$(\cdot)$ modules to advance the global feature representation, in which the $1 \times k$ convolution layer is served as the 1-order linear weight function followed by the Softmax function to evolve the attention

map. Besides, the value is also updated by injecting the integrated features at each stage. The whole procedure can be mathematically expressed as follows:

$$A_{i+1} = \text{Softmax}(A_i * C_i^1), \quad i = 1, 2, \cdots, N-1,$$
$$V_{i+1} = \text{Proj}(\text{Cat}(\mathcal{F}_i^g, \mathcal{F}_i^d)), \quad \mathcal{F}_{i+1}^g = A_{i+1} \otimes V_{i+1}, \tag{13}$$

where N is the number of the designed **LFormer**$(\cdot)$ module, Cat$(\cdot)$ denotes concatenation operation along the channel dimension, $*$ denotes the convolution operation, $C_i^1$ and $A_i$ represent the 1-order linear weight function and the evolved attention map with respect to the $i$-th **LFormer**$(\cdot)$ module, while $V_i$ is the corresponding value. $\mathcal{F}_i^g$ is the output of the $i$-th **LFormer**$(\cdot)$ module and $\mathcal{F}_i^d$ is detailed as below.
**Flow of the Feature Integration Branch.** We combine the high-frequency information and the output of the **LFormer**$(\cdot)$ module at each stage to update the value. Specifically, we first employ the Sobel operator to extract the high-frequency components of MS and PAN, denoted as $\widetilde{\mathcal{M}}$ and $\widetilde{\mathcal{P}}$, and then adopt several convolution blocks similar to those of the **LFormer**$(\cdot)$ branch to project them into shallow features. Then, we integrate the output of the **LFormer**$(\cdot)$ module and the high-frequency information through several convolution layers to update the value. Mathematically, this process is formulated as follows:

$$\widetilde{\mathcal{M}}, \widetilde{\mathcal{P}} = \text{Sobel}(\mathcal{M}, \mathcal{P}),$$
$$\mathcal{F}_0^d = \text{Proj}(\text{Cat}(\widetilde{\mathcal{M}}, \widetilde{\mathcal{P}})), \tag{14}$$
$$\mathcal{F}_i^d = \text{FIB}(\mathcal{F}_i^g, \mathcal{F}_{i-1}^d), \quad i = 1, 2, \cdots, N-1,$$

where Sobel$(\cdot)$ represents the Sobel operator, $\mathcal{F}_0^d$ is the extracted high-frequency low-level features, FIB$(\cdot)$ denotes fusing global and detail formation that is further used to update the value.

## 3.4 Loss Function

We adopt two loss terms, including the reconstruction loss $\mathcal{L}_r$ and the structure loss $\mathcal{L}_s$, as following:

$$\mathcal{L}_{total} = \mathcal{L}_r + \alpha \mathcal{L}_s, \tag{15}$$

where $\alpha$ is the hyperparameter that is used to balance the overall performance and the structure details. Specifically, we choose a widely used L1 loss to calculate the reconstruction loss $\mathcal{L}_r$, while the structure loss $\mathcal{L}_s$ is obtained through structural similarity (SSIM). They can be defined as follows:

$$\mathcal{L}_r = \|\mathcal{H}_s - \mathcal{GT}\|_1, \tag{16}$$
$$\mathcal{L}_s = \|1 - SSIM(\mathcal{H}_s, \mathcal{GT})\|_1, \tag{17}$$

where $\mathcal{H}_s$ and $\mathcal{GT}$ are the fused result and the matching ground truth, respectively.

## 4 Experiments

Datasets and experimental settings are detailed in supplementary.

### 4.1 Comparison with SOTAs

**Results on Reduced-resolution Scene.** We perform reduced-resolution assessment on WorldView-3 (WV3) and GaoFen-2 (GF2) datasets to quantitatively evaluate the similarity between the fused multispectral images and the ground truth images (original MS

**Table 1: Quantitative comparison between our model and state-of-the-art methods on reduced-resolution (Bold: best; Underline: second best).**

| Method | WV3 | | | | GF2 | | | |
|---|---|---|---|---|---|---|---|---|
| | SAM(± std)↓ | ERGAS(± std)↓ | Q8(± std)↑ | PSNR(± std)↑ | SAM(± std)↓ | ERGAS(± std)↓ | Q4(± std)↑ | PSNR(± std)↑ |
| BDSD-PC [28] | 5.4293±1.8230 | 4.6976±1.6173 | 0.8294±0.0968 | 32.9690±2.7840 | 1.6813±0.3596 | 1.6667±0.4453 | 0.8922±0.0347 | 35.1800±2.3173 |
| MTF-GLP-FS [29] | 5.3162±1.7663 | 4.7004±1.5966 | 0.8333±0.0923 | 32.9625±2.7530 | 1.6554±0.3852 | 1.5889±0.3949 | 0.8967±0.0347 | 35.5396±2.1245 |
| BT-H [19] | 4.9198±1.4252 | 4.5789±1.4955 | 0.8324±0.0942 | 33.0796±2.8799 | 1.6488±0.3603 | 1.5280±0.4093 | 0.9177±0.0253 | 36.0541±2.2360 |
| PNN [21] | 3.6798±0.7625 | 2.6819±0.6475 | 0.8929±0.0923 | 37.3093±2.6467 | 1.0477±0.2264 | 1.0572±0.2355 | 0.9604±0.0100 | 39.0712±2.2927 |
| DiCNN [9] | 3.5929±0.7623 | 2.6733±0.6627 | 0.9004±0.0871 | 37.3865±2.7634 | 1.0525±0.2310 | 1.0812±0.2510 | 0.9594±0.0101 | 38.9060±2.3836 |
| MSDCNN [37] | 3.7773±0.8032 | 2.7608±0.6884 | 0.8900±0.0900 | 37.0653±2.6888 | 1.0472±0.2210 | 1.0413±0.2309 | 0.9612±0.0108 | 39.2216±2.2275 |
| FusionNet [4] | 3.3252±0.6978 | 2.4666±0.6446 | 0.9044±0.0904 | 38.0424±2.5921 | 0.9735±0.2117 | 0.9878±0.2222 | 0.9641±0.0093 | 39.6386±2.2701 |
| DCFNet [31] | 3.0264±0.7397 | **2.1588±0.4563** | 0.9051±0.0881 | 38.1166±3.6167 | 0.8896±0.1577 | 0.8061±0.1369 | 0.9727±0.0100 | 40.2899±5.2718 |
| SFIIN [42] | 3.1004±0.6208 | 2.2499±0.5558 | 0.9105±0.0915 | 38.7768±2.8346 | 0.9275±0.1603 | 0.7914±0.1261 | 0.9733±0.0149 | 41.5664±1.5924 |
| PanViT [23] | 3.0923±0.6274 | 2.3329±0.6102 | 0.9053±0.0997 | 38.4300±2.9946 | 0.8066±0.1413 | 0.6998±0.1130 | 0.9783±0.0105 | 42.4268±1.6296 |
| InvFormer [40] | 3.2174±0.7010 | 2.3604±0.5774 | 0.9117±0.0863 | 38.3129±2.9141 | 0.7875±0.1497 | 0.6619±0.1178 | **0.9801±0.0085** | 42.8695±1.7598 |
| Fourmer [41] | 3.2363±0.6810 | 2.4189±0.6649 | 0.9108±0.0902 | 38.2682±2.7269 | 0.9757±0.2093 | 0.8845±0.1853 | 0.9698±0.0112 | 40.6700±1.9028 |
| LFormer | **2.8985±0.5835** | 2.1645±0.5089 | **0.9193±0.0861** | **39.0748±2.8440** | **0.6481±0.1299** | **0.5778±0.1123** | **0.9851±0.0067** | **44.1958±1.7995** |

**Table 2: Quantitative comparison between our model and state-of-the-art methods on full-resolution of GF2 dataset (Bold: best; Underline: second best).**

| Method | $D_\lambda$(± std)↓ | $D_s$(± std)↓ | HQNR(± std)↑ |
|---|---|---|---|
| BDSD-PC [28] | 0.0759±0.0301 | 0.1548±0.0280 | 0.7812±0.0409 |
| MTF-GLP-FS [29] | 0.0346±0.0137 | 0.1429±0.0282 | 0.8276±0.0348 |
| BT-H [19] | 0.0602±0.0252 | 0.1313±0.0193 | 0.8165±0.0305 |
| PNN [21] | 0.0317±0.0286 | 0.0943±0.0224 | 0.8771±0.0363 |
| DiCNN [9] | 0.0369±0.0132 | 0.0992±0.0131 | 0.8675±0.0163 |
| MSDCNN [37] | 0.0243±0.0133 | 0.0730±0.0093 | 0.9044±0.0126 |
| FusionNet [4] | 0.0350±0.0124 | 0.1013±0.0134 | 0.8673±0.0179 |
| DCFNet [31] | 0.0234±0.0116 | 0.0659±0.0096 | 0.9122±0.0119 |
| SFIIN [42] | 0.0418±0.0227 | 0.0666±0.0109 | 0.8943±0.0192 |
| PanViT [23] | 0.0304±0.0178 | 0.0507±0.0108 | 0.9203±0.0172 |
| Invformer [40] | 0.0609±0.0259 | 0.1096±0.0149 | 0.8360±0.0238 |
| Fourmer [41] | 0.0470±0.0391 | **0.0380±0.0097** | 0.9166±0.0352 |
| LFormer | **0.0206±0.0102** | 0.0501±0.0082 | **0.9303±0.0130** |

**Table 3: Average quantitative metrics on 11 examples for the CAVE ×4 dataset (Bold: best; Underline: second best).**

| Method | PSNR(± std)↑ | SSIM(± std)↑ | SAM(± std)↓ | ERGAS(± std)↓ |
|---|---|---|---|---|
| LTMR [5] | 36.5434±3.2995 | 0.9632±0.0208 | 6.7105±2.1934 | 5.3868±2.5286 |
| MTF-HS [30] | 37.6920±3.8528 | 0.9725±0.0158 | 5.3281±1.9119 | 4.5749±2.6605 |
| UTV [34] | 38.6153±4.0640 | 0.9410±0.0434 | 8.6488±3.3764 | 4.5189±2.8173 |
| ResTFNet [18] | 45.5842±5.4647 | 0.9938±0.0058 | 2.7643±0.6988 | 2.3134±2.4377 |
| SSRNet [39] | 48.6196±3.9182 | 0.9954±0.0024 | 2.5415±0.8369 | 1.6358±1.2191 |
| Fusformer [10] | 49.9831±8.0965 | 0.9943±0.0114 | 2.2033±0.8510 | 2.5337±5.3052 |
| HSRNet [11] | 50.3805±3.3802 | 0.9970±0.0015 | 2.2272±0.6575 | 1.2002±0.7506 |
| U2Net [25] | 50.4329±4.3655 | 0.9968±0.0023 | 2.1871±0.6219 | 1.2774±0.9732 |
| HyperTransformer [2] | 49.5532±3.1812 | 0.9967±0.0011 | 2.2323±0.6706 | 1.2587±0.7628 |
| DHIF [12] | 51.0721±4.1648 | 0.9973±0.0017 | **2.0080±0.6304** | 1.2216±0.9653 |
| LFormer | **51.5521±3.9542** | **0.9974±0.0013** | 2.0600±0.6095 | **1.0967±0.8256** |

exhibits the quantitative results of all compared pan-sharpening methods on GF2 datasets. Our proposed framework demonstrates superior performance again, showcasing its exceptional generalization capacity.

## 4.2 Extension to Hyperspectral Task

We extend our model to the application of hyperspectral image super-resolution (HISR). This application shares similar degradation principles with the multispectral pan-sharpening task. We compare our model with several state-of-the-art methods using the widely used CAVE dataset. Table 3 showcases our model's superior performance, surpassing the compared methods by a significant margin. Figure 8 displays the residual maps between the fused images and the GT image, and their spectral response at a spatial location [400, 200]. It is apparent that the dark blue residual map further demonstrates the high similarity between the fused product

images). Table 1 presents the average performance of all compared pan-sharpening methods on WV3 and GF2 datasets, respectively, where our model achieves the best results across all metrics. Figure 5 and 6 display the qualitative comparisons of the error maps between our model and other cutting-edge methods over WV3 and GF2 datasets. It is clearly observed that our model presents a favorable outcome evidenced by its dark blue residual map.

**Results on Real-world Full-resolution Scene.** To evaluate the generalization in real-world scenes, we conduct full-resolution evaluation. As ground truth images are not available, we rely on quality indexes without reference for performance assessment. Model trained on reduced-resolution data are applied to real scenes. Table 2

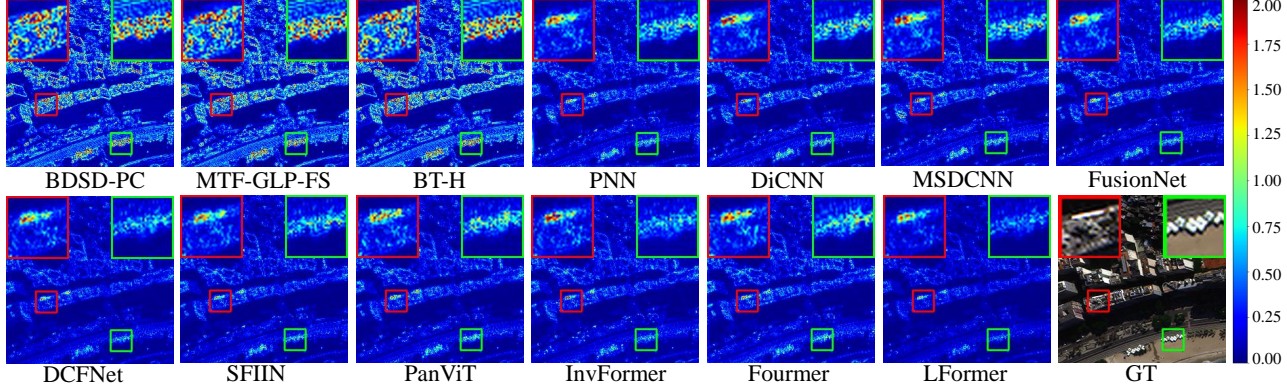

**Figure 5: Comparison of the error maps between our model and other cutting-edge methods over WV3 dataset.**

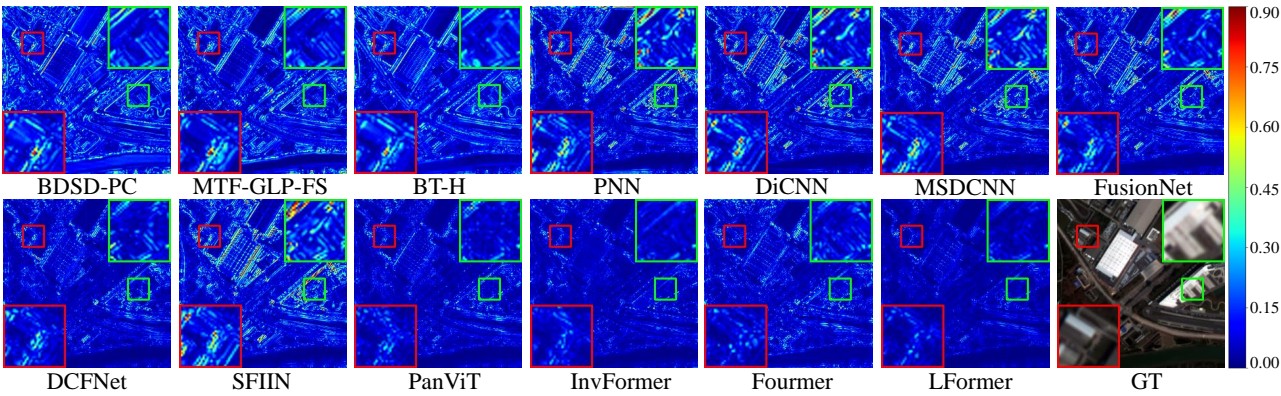

**Figure 6: Comparison of the error maps between our model and other cutting-edge methods over GF2 dataset.**

of our model and the GT image. Moreover, the spectral response curve of our model closely aligns with the GT, indicating its desired spectral preservation capability.

## 4.3 Visualization of Feature Maps

To further demonstrate the feature representation capabilities of our model, we provide the feature maps of different blocks as displayed in Figure 7. It is clearly that the feature map becomes more distinguishable and provides rich detail information with the increase of the number of blocks, thereby proving the desirable expressiveness of our proposed key linearly-evolved transformer design.

## 5 Ablation Study

All ablation experiments are implemented on GF2 dataset.

**Effect of Attention Evolution.** We use the proposed LFormer as the baseline and compare it with alternative methods by altering the approach for calculating the attention map. All comparison networks are trained using the same methodology. Specifically, we compare three configurations: **Baseline**: Conducting the cross-attention in the first block, followed by leveraging our core linear evolution strategy to evolve the attention weights in the remaining blocks. **Config.I**: Performing cross-attention computation at each

block. **Config.II**: Removing the linear evolution within the baseline, thereby directly sharing the cross-attention weights obtained from the first block with the remaining blocks. Table 4 presents the quantitative results of different models. Our proposed LFormer consistently achieves the best outcomes while significantly reducing the number of network parameters and FLOPs. It clearly illustrates the performance benefits of LFormer against its variants. In particular, the Config.I incorporates the repetitive and unnecessary self-attention computations, leading to increased model parameters and FLOPs while inferior performance.

**Effect of Kernel Size.** To examine the function of the employed 1-D convolution kernel within the evolved process, we select several representative 1-dimensional convolution units with $1\times1$, $1\times3$, $1\times5$, and $1 \times 7$ kernel size. From the reported quantitative comparison in Table 5 over the proposed LFormer and its variants on the GF2 dataset, it can be deduced that with the kernel size increasing, the model performance tends to improve. While there is a slight decrease in performance when the kernel size increases to $1 \times 7$, probably due to the attention weights between different layers experiencing local fluctuation only. Therefore, we set the $1 \times 5$ kernel size as default.

**Extendibility.** We further apply the proposed linear evolution paradigm to local attention mechanism to validate its scalability. Similar

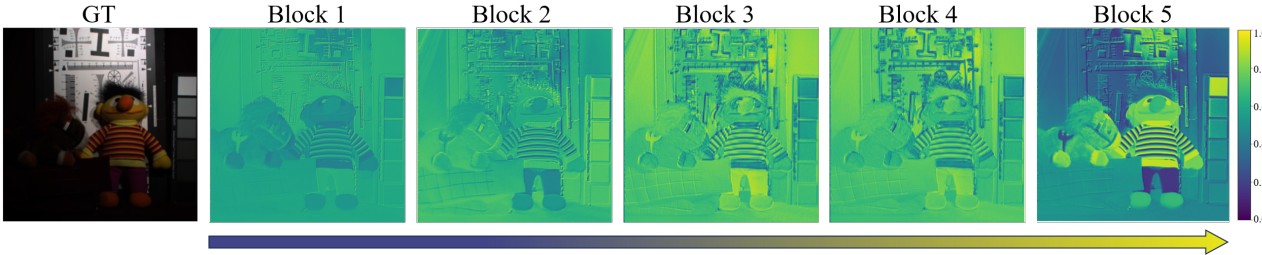

**Figure 7: Visualization of feature maps in different blocks.**

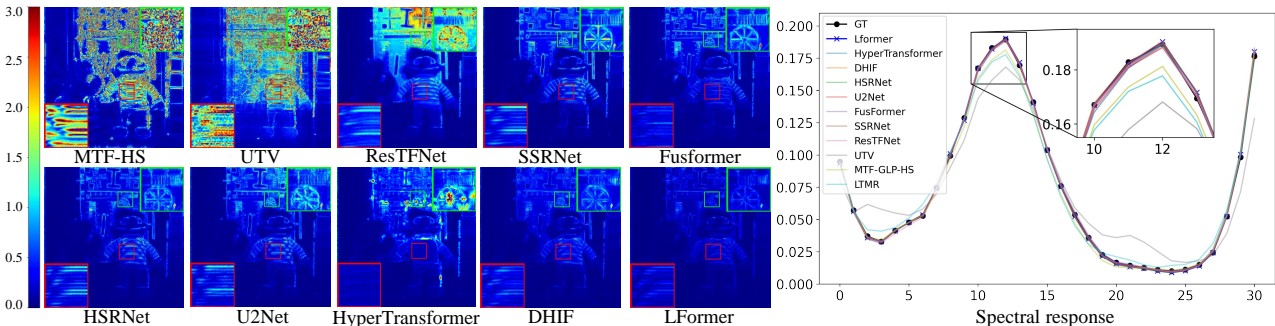

**Figure 8: Comparison of the error maps and spectral responses between our model and other methods on CAVE ×4 dataset.**

**Table 4: Quantitative comparison of LFormer and its variants.**

| Method | Reduced | | | | Params | FLOPs |
|---|---|---|---|---|---|---|
| | SAM | ERGAS | Q4 | PSNR | | |
| Config.I | 0.6523 | 0.5782 | 0.9847 | 43.9789 | 2.327M | 9.528G |
| Config.II | 0.7138 | 0.6316 | 0.9829 | 43.4172 | 0.588M | 2.380G |
| Baseline | 0.6481 | 0.5778 | 0.9851 | 44.1958 | 0.589M | 2.448G |

**Table 5: Quantitative comparison of different kernel size.**

| Kernel Size | Reduced | | | | Params | FLOPs |
|---|---|---|---|---|---|---|
| | SAM | ERGAS | Q4 | PSNR | | |
| $1 \times 1$ | 0.7338 | 0.6329 | 0.9813 | 43.2811 | 0.589M | 2.447G |
| $1 \times 3$ | 0.6817 | 0.6119 | 0.9837 | 43.6932 | 0.589M | 2.447G |
| $1 \times 5$ | 0.6481 | 0.5778 | 0.9851 | 44.1958 | 0.589M | 2.448G |
| $1 \times 7$ | 0.6632 | 0.5832 | 0.9840 | 43.8974 | 0.589M | 2.449G |

to the ablation experiment 1, we investigate three model configurations with different attention computation manners. **Baseline**: Conducting the window attention in the first block, followed by leveraging our core linear evolution strategy to evolve the attention weights in the remaining blocks. **Config.I**: Performing window attention computation at each block. **Config.II**: Removing the linear evolution within the baseline, thereby directly sharing the window attention weights obtained from the first block with the remaining blocks. As reported in Table 6, the model configured with our linear evolution strategy yields the best results despite the slight increments in parameters and FLOPs compared to Config.II, showcasing its promising applicability.

**Table 6: Quantitative comparison of different variants by extending our linear evolution strategy to window attention.**

| Method | Reduced | | | | Params | FLOPs |
|---|---|---|---|---|---|---|
| | SAM | ERGAS | Q4 | PSNR | | |
| Config.I | 0.7919 | 0.7079 | 0.9799 | 42.4992 | 2.103M | 8.769G |
| Config.II | 0.7721 | 0.6603 | 0.9812 | 42.8917 | 0.571M | 2.427G |
| Baseline | 0.7167 | 0.6487 | 0.9827 | 43.0769 | 0.573M | 2.649G |

## 6 Limitation

We assess the effectiveness of our proposed framework in panchromatic and multispectral image fusion, as well as hyperspectral image fusion task. Additionally, we aim to test the scalability and versatility of the core linearly-evolved transformer in other low-resource image restoration tasks, such as efficient image super-resolution and Ultra-High-Definition tasks. To highlight, our primary objective is to offer an alternative global modeling framework with an efficient structure.

## 7 Conclusion

We propose an efficient variant of the linearly-evolved transformer for lightweight pan-sharpening. By interpreting self-attention as a 1-order linear weight function, we replace the N-cascaded transformer chain with a single transformer and N-1 convolutions. Leveraging this insight, we develop an alternative 1-order linearly-evolved transformer using 1-dimensional convolutions. Extensive experiments on multispectral and hyperspectral image sharpening tasks confirm the competitive performance of our method against state-of-the-art approaches.

## Acknowledgments

This work was supported by the National Natural Science Foundation of China under Grant 42241109 and Grant 42271350.

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
