# OpenReview forum: "Linearly-evolved Transformer for Pan-sharpening"
_acmmm.org/ACMMM/2024/Conference — MM2024 Poster_

### Official Review · Reviewer_Zies · 2024-05-23

**Rating:** 4
**Confidence:** 3

**Summary:**

For the pan-sharpening task, this work proposes an efficient linearly evolved transformer variant that replaces cascaded self-attention with the 1-dimensional linear convolution chain for computational efficiency.

**Strengths:**

The paper is sound, clear motivation, well-organized, and articulately presented, ensuring it is easily understandable.

**Limitations:**

- This paper focuses on lightweight structures, therefore, it is suggested that parameters and computations are listed in Table I for intuitive comparison.

- Since a 1-dimensional linear convolution chain is used instead of a cascaded SA. Therefore, Figure 7 should give a visualisation of the features in comparison with the transformer method.

- What is the role of extracting high-frequency boundary information? It is suggested to show its impact in ablation experiments.

- typo: L565, perfrm?

**Suitability:**

2

---

### Official Review · Reviewer_x2cE · 2024-05-23

**Rating:** 4
**Confidence:** 3

**Summary:**

In this paper, the authors propose an efficient variant of the linearly-evolved transformer for lightweight pan-sharpening. By interpreting self-attention as a 1-order linear weight function, the authors replace the N-cascaded transformer chain with a single transformer and N-1 convolutions.

**Strengths:**

The authors delve into the popular cascaded transformer modeling with cutting-edge methods and develop an alternative 1-order linearly-evolved transformer variant with a 1-dimensional linear convolution chain to achieve the same function. This approach allows their proposed method to benefit from the cascaded modeling rule while achieving favorable performance in an efficient manner.

**Limitations:**

1.The author should strengthen the article's motivation by conducting a detailed analysis of why current mainstream transformer models use a cascading approach to stack transformer blocks.
2.With rapid advancements in the field of Pan-sharpening, it is strongly advised that the authors compare their proposed method with more recent algorithms published within the last two years, as some of the comparison algorithms used in the study may be outdated.
3.Figure 3 should include a comparison of this article's approach with ViT.
4.The model's feature representation capability is illustrated in Figure 7. The author should elaborate on why replacing the n-cascade transformer chain with a single transformer and N-1 convolution can enhance the feature representation capability of the model.

**Suitability:**

2

---

### Official Review · Reviewer_AHWp · 2024-05-25

**Rating:** 4
**Confidence:** 4

**Summary:**

This study focuses on the pan-sharpening problem and introduces a modified transformer model to alleviate the computational burden associated with the traditional transformer architecture. The proposed approach achieves efficient computation and competitive performance by introducing a 1-order linearly-evolved transformer variant with a 1-dimensional linear convolution chain. Experiments conducted on widely-used satellite images validate the effectiveness of the proposed method.

**Strengths:**

The idea of linearly-evolved attention is interesting, and the experiments showcase the effectiveness of the proposed method.

**Limitations:**

1. The paper suggests that the newly proposed linearly-evolved attention reduces computation costs. However, it only provides the FLOPs, and it's unclear how the computation complexity of the proposed attention compares to the O(N^2) complexity of traditional self-attention and the running speed of the method.
2. The paper adopts the fancy name linear-evolved attention, where its core operator is $1\times N$ convolutions. One-dimensional convolution has been used widely in the design of light-weight CNN networks, such as InceptionNet. The novelty of the proposed attention is questionable.

**Suitability:**

3

---

### Official Review · Reviewer_yL8S · 2024-05-26

**Rating:** 3
**Confidence:** 3

**Summary:**

In this paper, a novel efficient variant of the linearly-evolved transformer is proposed for lightweight pan-sharpening. More specifically, the N-cascaded transformer chain is replaced by a single transformer and N-1 convolutions and the Soble operator is adopted by high-frequency information extraction.

**Strengths:**

1. The performance of this work is good.
2. The motivation is somewhat novel. In this paper, they regard the cascaded transformer chain as the linearly-evolved transformer. From this perspective, designing a lightweight model is interesting.
3. The legend and experiments are clear.

**Limitations:**

1. The article is readable, but somewhat difficult to understand, and a more organized presentation would be helpful.
2. The clarify in “Proposed Method” section is not clear in detail. How about the details of “Proj” operation? “FIB” means the fusion of global and detail formation, but how to fusion? And where the Soble operator employ?  Maybe illustrated in framework figure 4 better.
3. There are also some efficient linear attention modules based on ViT proposed in image classification, such as Flatten Transformer[1] in ICCV 2023. The differences between this method and these methods should be discussed, and the linear chains can be replaced by these linear attention?
[1]. Han, Dongchen, et al. "Flatten transformer: Vision transformer using focused linear attention." Proceedings of the IEEE/CVF International Conference on Computer Vision. 2023.
4. The experiments validation is lacking in some aspects and the effectiveness of methods in this paper can not be convinced.
-In ablation study, the effectiveness of sobel operator and feature integration should be validated.
-The FLOPs, Params, Latency metrics about efficiency is lack in table 1 and table 2.

**Suitability:**

2

---

### Meta-Review · Area_Chair_6i6d · 2024-06-29

**Recommendation:** Accept (Poster)
**Confidence:** 4

**Metareview:**

This work develops an efficient linearly-evolved transformer variant to construct a lightweight framework to address the challenge of high model parameters and FLOPs in satellite pan-sharpening using vision transformer family, achieving competitive performance with fewer resources and verified in hyper-spectral image fusion. Reviewers raised concerns about the novelty of the approach, unclear method description and insufficient experiments. With the authors' efforts in the rebuttal, most of the concerns are resolved. The authors are encouraged to improve the paper based on the reviewers' feedback.